# Past and future trends in fire weather for the UK

Matthew C. Perry[1], Emilie Vanvyve[1], Richard A. Betts[1,2], Erika J. Palin[1]

[1]Met Office, FitzRoy Road, Exeter, EX1 3PB, UK
[2]Global Systems Institute, University of Exeter, Laver Building, North Park Road, Exeter, EX4 4QE, UK

5 *Correspondence to*: Matthew C. Perry (matthew.perry@metoffice.gov.uk)

**Abstract.** Past and future trends in the frequency of high danger fire weather conditions have been analysed for the UK. An analysis of satellite-derived burned area data from the last 18 years has identified the seasonal cycle with a peak in spring and a secondary peak in summer, the high level of interannual variability, and the lack of a significant trend despite some large events occurring in the last few years. These results were confirmed with a longer series of fire weather indices back to 1979. The Initial Spread Index (ISI) has been used for spring, as this reflects the moisture of fine fuel surface vegetation, whereas conditions conducive to summer wildfires are hot, dry weather reflected in the moisture of deeper organic layers which is encompassed in the Fire Weather Index (FWI).

Future projections are assessed using an ensemble of regional climate models from the UK Climate Projections, combining variables to derive the fire weather indices. The results show a large increase in hazardous fire weather conditions in summer. At 2°C global warming relative to 1850-1900, the frequency of days with 'very high' fire danger is projected to double compared to a recent historical period. This frequency increases by 5 times at 4°C of global warming. Smaller increases are projected for spring, with a 150% increase for England at 2°C of global warming and a doubling at 4°C. A particularly large projected increase for late summer and early autumn suggests a possible extension of the wildfire season, depending on fuel availability.

These results suggest that wildfire can be considered an 'emergent risk' for the UK, as past events have not had widespread major impacts, but this could change in future, with adaptation actions being required to manage the future risk. The large increase in risk between the 2°C and 4°C levels of global warming highlights the importance of global efforts to keep warming below 2°C.

## 1 Introduction

In recent years, large wildfires have occurred across many parts of the world. Areas affected have included Australia, California, Amazonia, Siberia and southern Europe. Many of these events have had severe impacts, such as loss of human lives, loss of wildlife and their habitats, destruction of houses and buildings and degraded air quality (Deb et al., 2020; Wang et al., 2021), and this has highlighted the threat posed by this natural hazard. In the UK, impacts have so far tended to be less severe and threats to human life are rare, but major incidents have led to significant impacts on air quality, wildlife, ecology and infrastructure (Belcher et al., 2021). Irrecoverable damage to peat soils is a particular concern as they represent a significant store of carbon, especially in Scotland (Milne and Brown, 1997). The suppression of wildfires by fire and rescue services is also costly and resource intensive.

### 1.1 Wildfire occurrence in the UK

In the UK, severe wildfire is an intermittent hazard as the most serious incidents are concentrated in a few dry years. Wildfire incidents are most prevalent in the spring because of the availability of dead and dry fine vegetation as fuel, but widespread wildfires have also occurred in some hot, dry summers. The UK Fire and Rescue Services attend around 32,000 wildfires each year (Forestry Commission, 2019). However, the vast majority of wildfires are small incidents – Gazzard et al. (2016) reported that 99% of Great Britain fires affect less than one hectare. Most wildfires occur at the rural-urban interface or on arable land, as this is where fires are most likely to be ignited by human activity. However, many of the larger wildfires occur in more remote areas, especially moorland, forests, and peatland bog. In these areas, suppression is more difficult, and there is a greater continuity of fuel available. Forest and woodland fires currently constitute a relatively small fraction of all wildfire incidents, but their impacts can be large and costly.

For example, the Swinley Forest fire in April/May 2011 damaged 110 ha of habitat and was particularly resource intensive for the fire and rescue service, occurring near to residential areas in Berkshire, southeast England (Brown et al., 2016). Another event which occurred recently was the Wareham Forest fire, which affected 200 hectares in Dorset in May 2020 (Belcher et al., 2021). As yet, forest fires have not impacted directly on residential properties in the UK, but with a number of settlements being located in or near forested land, there is a level of exposure to wildfire should fires occur in these locations. After the severe fire season in 2011, severe wildfire was added to the National Risk Register in 2013 (Gazzard et al., 2016).

### 1.2 Drivers of wildfire activity

Weather conditions have a direct impact on the likelihood and severity of fires occurring. Variations in weather such as temperature, relative humidity and precipitation affect the amount of moisture held in both live and dead vegetation ('fuel') – which is crucial for flammability. As well as short-term conditions, soil moisture is affected by antecedent rainfall amounts over previous weeks and months. Fuel availability is higher in the spring and summer than in the autumn and winter. In

spring, a large amount of ground vegetation is dry (not in a growth phase yet) or dead, and acts as fuel. Fires in spring are generally fuelled by dead grasses, heather and surface vegetation litter. These fuels become very dry during warm weather when the relative humidity is low (50% or less). This typically occurs when high pressure dominates the UK or the wind direction is from the east, bringing dry continental air to the UK. Wind speed also influences fuel moisture but is more

closely related to the behaviour of fire, favouring spread. Strong winds help fires to spread more quickly and there is usually insufficient green vegetation in spring to prevent the spread of a fire. In summer, however, fires are generally driven by high temperatures and prolonged dry conditions which reduce the amount of moisture in the living vegetation. Both seasons exhibit spells of dry, warm/hot, windy weather which therefore favours the start and spread of wildfires, but conditions can fluctuate greatly on a daily basis. More generally, weather also impacts on vegetation growth and thereby future fuel

availability. Spatial variations in climate are also an important factor for the type and structure of vegetation.

As well as having suitable conditions to spread, wildfires need an ignition source. Lightning is a weather event that can cause wildfires to ignite, particularly in the case of 'dry lightning' (not accompanied by significant rainfall) (Read et al., 2018). This is a significant source of ignitions in some parts of the world, for example Australia and Canada, but is very rare in the UK where the vast majority of wildfires are started by human activity, either accidentally or deliberately.

In addition to the weather, the occurrence and spread of wildfire are also affected by human factors including accidental or deliberate ignition, land use, land management and any efforts to suppress the fire. Land management such as clearing of vegetation and prescribed burning can reduce fuel availability. In terms of a risk framework, these human factors can be considered to affect the vulnerability and resilience of an area to the wildfire hazard. The risk, or possibility of adverse impacts, is also affected by the value of natural and infrastructure assets exposed to the wildfire hazard.

**1.3 Fire danger**

Fire-danger indices are a long-standing attempt at combining wildfire-related weather information into a value which represents the danger that a wildfire would pose, should one be ignited. The weather variables most commonly used in the calculation of these indices are air temperature, relative humidity, wind speed and precipitation. Various such indices exist and are in use worldwide, for example the Canadian Fire Weather Index (van Wagner, 1987) and the McArthur Forest-Fire-

80 Danger Index (Noble et al., 1980). Other indices add information on fuel and topography to provide a more complete picture of fire danger, such as the Fire Potential Index which include satellite-derived vegetation type and greenness and terrain data (Burgan et al., 1998).

This study uses the Canadian Fire Weather Index (FWI) because we focus on the impacts on climate change, which affects fire danger mainly through weather, and it is also used by the Met Office for a daily forecast service. The FWI provides a

85 quantification of how favourable conditions are for a grass or forest fire to spread and intensify, should one have been started. The model comprises six indices tracking fuel moisture content, the rate of spread and fire intensity. Calculations are based on consecutive daily observations of air temperature, relative humidity, wind speed and 24-hour rainfall taken at noon, but the resulting indices are representative of the mid-afternoon-peak fire danger (van Wagner, 1987). Rainfall is also

assessed over a period of several prior months to estimate the moisture content of the vegetation and soil layers. See Section 2.2 for further details.

## 1.4 Wildfire and climate change

Several studies have established links between climate change and wildfire internationally. Climate trends can increase the likelihood of severe fire weather through increases in average temperature and in the frequency, intensity, duration and/or extent of heatwaves and droughts. Fire weather seasons have significantly lengthened across 25% of the Earth's vegetated surface (Smith et al., 2020). The impact of anthropogenic climate change on fire weather has emerged above natural variability for Amazonia, Southern Europe/Mediterranean, Scandinavia, the Western USA and Canada (Abatzoglou et al., 2019). These impacts will intensify with further warming, and future increases in fire weather risk are projected for many areas, including the Iberian Peninsula (Calheiros et al, 2021) and Canada (Wang et al., 2017). The response of lightning to climate change is uncertain, but recent research shows no significant change in lightning flash rates for the UK, with a projected decrease globally (Finney et al., 2018).

The occurrence and spread of wildfires are also influenced by human factors such as ignitions, land management and fire suppression (Wu et al., 2021). At the global scale, burned area has been found to decrease in recent decades, mainly due to agricultural expansion in savannas (Andela et al., 2017). However, in developed nations such as the UK which are already heavily urbanised, human factors are unlikely to change significantly in future, so climate is likely to remain the main factor limiting future changes (Wu et al., 2021).

Few studies have looked in detail at projected changes for the UK. Some global studies include mapped results for the UK, for example Abatzoglou et al. (2019) project climate change impacts to emerge in southern Britain at a global warming level of 2–3°C. A review of evidence for the second UK Climate Change Risk Assessment (CCRA2) found with medium confidence that projections of drier summers with increased soil moisture deficits would suggest an increase in the number of fires and the area affected (Brown et al., 2016). They also reported greater uncertainty for changes in spring. Albertson at al. (2010) simulated local weather for the Peak District in northern England, finding a projected increase in summer wildfires after 2040. The latest set of climate projections for the UK is provided by the UK Climate Projections. The headline findings are for warmer, wetter winters and hotter, drier summers in the future (Lowe et al., 2018). Arnell et al. (2021) used 60 km resolution projections from UKCP18 and projected increases in the occurrence of high fire danger for most regions of the UK using a high-emissions scenario, but very limited change with a low-emissions scenario. Warmer, wetter winters could increase fuel load due to greater vegetation growth.

## 1.5 Aims of this study

The wildfire hazard is represented in this study by the FWI and its sub-indices exceeding thresholds relevant for UK conditions in spring and summer. This study focuses on projected changes in the frequency of days with very high fire danger for global warming levels of 2°C and 4°C. This aims to compare possible outcomes if the world is successful in

meeting targets set by the Paris Agreement in 2015 to keep the increase in global average temperature well below 2°C above pre-industrial levels, with potential outcomes if these targets are not met. Future changes are also considered in the context of past trends in fire danger and burned area.

Section 2 of this article describes the approach taken to assess past wildfire events, the type of fire-danger indices used in the analysis and how they have been calculated for past and future periods. Section 3 looks at past occurrences of moderate to severe wildfires using MODIS and EFFIS satellite-based burned areas, as well as past fire danger using the fire-danger indices calculated from the ERA5 reanalysis. Section 4 looks at future projections of fire danger in a changing climate. Section 5 discusses the results and concludes.

## 2 Methods and data used

### 2.1 Satellite-derived fire observations

Satellites are a useful tool to detect large-enough wildfires and to provide an insight into their footprint (burned area). Satellites detect wildfires through increased thermal emissions and changes in vegetation-related reflectance caused by burn scars (Humber et al., 2018). Algorithms use either or both of these to estimate the date and burned area of wildfires. Uncertainties can be considerable but are usually mitigated by the algorithms. Uncertainties typically stem from restrictions of satellite views (caused by cloud cover), spatial and temporal coverage of satellite overpasses, and non-fire-related vegetation changes such as cropland harvest or forest clearing (Boschetti et al., 2019).

Several wildfire-relevant satellite products make use of sensors of the Moderate Resolution Imaging Spectroradiometer (MODIS) on board the Terra and Aqua satellites, operated from the USA by the National Aeronautics and Space Administration. The Collection 6 product MCD64A1 (Giglio et al., 2018) provides global burned areas on a grid of 0.004 x 0.004 degrees of latitude and longitude – about 260 m x 460 m spatial resolution for the UK – and usually has a twice-daily overpass. Validation of the product has found an omission error of 73% and a commission error of 40%, with approximately half the overall area mapped compared to the reference data (Boschetti et al., 2019). 90% of the area difference is due to 3 km grid cells with less than 25% of their area burned, as small and spatially fragmented burned areas are not mapped at the 500 m scale at which the MCD64A1 product was calculated.

These data have been extracted for the UK and the total monthly burned area across the UK calculated. Data go back to 2003 and provide a 17-year-long time series of burned areas. This time series is too short for a robust assessment of trends, but it can provide useful information on monthly and inter-annual variability and an indication of recent changes. In addition to the MODIS burned area, the burned-area dataset of the European Forest Fire Information System (EFFIS) was used, as a point of comparison. This dataset is also derived from the daily processing of MODIS satellite imagery. It is available from 2008 onwards and represents 75 to 80 % of the total burned area in Europe by fires larger than 30 hectares in size.

## 2.2 Fire Weather Index

The Met Office provides a weather-related fire-danger forecast for emergency planners in England and Wales. During the implementation of the system, several models were examined, and the choice was made to use the Canadian FWI system for modelling fire danger in the UK because it was found suitable to pick out periods of greater fire danger in wet and hot, dry summers as well as in spring (Met Office, 2005). It has also been in use since 2007 in the European service EFFIS.

The FWI is comprised of five sub-indices. The Initial Spread Index (ISI) is a numeric rating of the expected rate of fire spread. It combines the effects of wind and fine fuel moisture (the Fine Fuel Moisture Code, FFMC) on rate of spread without the influence of variable quantities of fuel. The Build-up Index (BUI) is a numeric rating of the amount of fuel available for combustion, which combines temperature, rainfall and relative humidity to represent the moisture content of organic layers – the Duff Moisture Code (DMC) for moderate depth layers and the Drought Code (DC) for deep layers. The ISI and BUI are then combined to arrive at the overall FWI.

In order to provide a fire-danger rating, the FWI is divided into five classes ranging from 1 (low danger) to 5 (exceptional danger). The thresholds defining each class (or level of fire danger) were adapted for the UK (Kitchen, 2010), considering the frequency of values historically experienced across a selection of sites, and following the method of van Wagner (1987) based on progressive values of a fire intensity scale related to the FWI. The thresholds are shown in Table 1. The UK 'exceptional danger' class was specifically designed to be very rare and is used to allow closure of public-access land. These levels are applied uniformly across the whole of the UK, which might not be reflective of the actual fire risk in each part of the country. For example, lower thresholds may be more appropriate for conditions in Scotland, with fires able to ignite above an ISI of 2. Vulnerability to wildfire is also likely to vary considerably across the country due to factors such as land use and vegetation type. Note that EFFIS uses alternative thresholds more suitable for southern European conditions.

De Jong et al. (2016) evaluated each of the FWI components during UK wildfire events, finding that the FFMC and ISI sub-indices by themselves performed best in spring, and that the overall FWI showed the greatest skill in summer. This is because most spring wildfires require dry fine fuels to allow self-sustaining ignitions, as the moisture content of vegetation is generally lower than in summer due to limited growth. In summer however, most wildfires tend to occur during prolonged dry periods, so the BUI is also important, leading to the overall FWI performing best. Similar results were obtained by Davies and Legg (2016) for Scotland, except that here the cooler, wetter climate means that drought conditions are unusual, which limits the effectiveness of the FWI in summer The Natural Hazards Partnership Daily Hazard Assessment (Hemingway and Gunawan, 2018), an overview of natural hazards that could affect the UK over the next 5 days which is provided to emergency planners and responders, uses forecast exceedances of ISI 'high' danger level to warn of elevated fire-weather conditions from November to May and exceedances of FWI 'very high' danger level from June to October. This approach has also been taken for the results presented in this report.

**Table 1: Fire-danger classes used by the Met Office for the UK (in particular England and Wales).**

| Level of fire danger | FFMC | ISI | BUI | FWI |
|---|---|---|---|---|
| 1 (Low) | < 63 | < 2 | < 20 | < 4.54 |
| 2 (Moderate) | 63 – 84 | 2 – 5 | 20 – 30 | 4.54 – 9.38 |
| 3 (High) | 84 – 88 | 5 – 10 | 30 – 40 | 9.38 – 17.35 |
| 4 (Very high) | 88 – 91 | 10 – 15 | 40 – 60 | 17.35 – 52.36 |
| 5 (Exceptional) | > 91 | > 15 | > 60 | > 52.36 |

## 2.3 Historical weather data

Historical daily values of the FWI and its sub-indices have been calculated by the Copernicus Emergency Management Service using ERA5 reanalysis data (Vitolo et al., 2020). This readily available dataset was extracted from the Climate Data Store for the period 1979 to 2020 (Copernicus, 2019) and was used to provide a good proxy for observations of past fire-weather conditions. Gridded reanalysis datasets provide a spatially complete and consistent record of the global atmospheric circulation, incorporating observed data from a range of sources. ERA5 is the latest generation reanalysis dataset produced by the European Centre for Medium-Range Weather Forecasts (ECMWF) (Hersbach et al., 2020). Its spatial resolution is 0.25° in latitude and longitude (approximately 28 x 15 km over the UK).

## 2.4 UKCP18 climate model scenarios

Future scenarios of climate change are based on outputs from one of the components of the UK Climate Projections 2018 (UKCP18). This component uses an up-to-date climate model downscaled to a 12 km scale over the UK to assess how the climate may change in response to increasing radiative forcing (Met Office Hadley Centre, 2018). These projections comprise a perturbed parameter ensemble of 12 members (PPE-12) of the Met Office Hadley Centre regional climate model HadREM3-GA705 (Fung et al., 2018), providing a range of plausible outcomes that can reveal the potential for extreme conditions. However, these regional projections are downscaled from a perturbed-parameter ensemble of the Met Office Hadley Centre global climate model HadGEM3-GC3.05 (Fung et al., 2018), and as such they do not cover the full range of uncertainty. In addition, HadGEM3-GC3.05 samples the warmer end of the warming climate response – the model has a higher climate sensitivity to greenhouse gases (Lowe et al., 2018).

We examine projected changes in fire weather global warming levels (GWLs) of 2°C and 4°C above 1850–1900, since these levels of warming are routinely examined in the UK Climate Change Risk Assessment (Committee on Climate Change, 2017). 4°C global warming by 2100 is within the range of possible outcomes for emissions scenarios consistent with a continuation of current policies (Betts, 2020). We obtain projected regional climate changes at these GWLs from projections driven by the RCP8.5 emissions scenario (Murphy et al., 2019). This is a scenario of very high emissions, above those

considered consistent with a continuation of current worldwide policies and is used to produce high-end climate change scenarios for use in risk assessments – it is not intended to be interpreted as a projection of the most likely outcomes.

Projections with this scenario reach GWLs of 2°C and 4°C earlier than would be reached with emissions scenarios consistent with current policies. However, for quantities known to scale linearly with global warming levels, such as many aspects of extreme weather (Wartenberger et al., 2017), it can be appropriate to treat such changes as representative of the regional climate state reached at the same level of warming at later times (Bärring and Strandberg, 2018). Periods corresponding to global warming levels of 2°C and 4°C above 1850–1900 were identified in the RCP8.5 12 km projections and are used as a

proxy for the future wildfire hazard at these levels which may then be translated to other pathways of greenhouse-gas emissions. For each ensemble member, an 11-year period centred on the year when the model run first reached the relevant global warming level was used, as shown in Table 2 of the UKCP18 report on derived projections (Gohar et al., 2018). For the 2°C warming level, these central years range from 2027 to 2035, and for the 4°C level from 2057 to 2070 across the ensemble.

Modelled past and future fire danger was calculated with a Python 3 implementation of the Canadian FWI system (Wang et al., 2015) with the relevant fire-danger classes for application to the UK (Table 1). Input weather variables from the regional UKCP18 datasets were daily values of maximum air temperature, mean relative humidity, mean wind speed and precipitation. These were used as a proxy for noon values, with precipitation totals from noon to noon being estimated by averaging two daily values. The use of daily values over actual noon values likely introduces a slight bias in the results, but

this is mitigated by using the daily maximum temperature over the daily mean temperature, as the former is usually close to the noon value.

All climate models exhibit differences between modelled results and observations. Calibration techniques, such as bias adjustment, can be used to account for systematic differences. However, bias adjustment of multivariate indices such as the FWI is complex, and there are issues both with applying techniques directly to the index and with separately adjusting each

230 individual weather parameter (Casanueva et al, 2018). Therefore, calibration techniques have not been applied to the results and the model-derived results shown should be interpreted with care. While future trends such as an increasing occurrence of severe fire danger are likely to be robust, absolute numbers such as percentages of days exceeding certain threshold values would be likely to change if a recalibration technique is applied to the model data.

## 3 Past trends in UK wildfire occurrence

There are no long-enough records on the occurrence of wildfires in the UK to allow for the estimation of trends. Wildfire incidents in England have been recorded in the Fire & Rescue Services Incident Recording System since 2009 only. Therefore, past trends have been assessed using satellite-derived data on burned area and fire weather indices.

### 3.1 Burned area

The monthly UK burned areas recorded by the MODIS satellite from 2003 to 2019 are shown in Fig.1 and the corresponding annual cycle of the multiple-year mean burned area for each month of the year in Fig.2. These show that the greatest total burned area has occurred in spring, with a peak in April. There is also a significant total for February, mainly due to events in 2003, although there have also been February wildfires in recent years (BBC News, 2019a). The annual cycle shows a secondary peak in the summer around the month of July. The three largest multi-month burned areas detected by the MODIS satellite over the last 17 years are indeed respectively from the springs of 2019, 2003 and 2007. These springs were all characterised by warm and dry conditions conducive to wildfires, leading to events which had significant impacts. In the spring of 2019, Scotland was hit by several wildfires, two of which were particularly large (BBC News, 2019b). The fire in Moray in April burned approximately 50 km$^2$ of grassland and threatened a wind farm. The fire in Sutherland in May burned 80 km$^2$ of moorland (heather, grass and dry peatland) including part of the Forsinard Flows RSPB natural reserve and threatened electricity supply to 800 homes. In 2003, several fires occurred in the Peak District in April; the Bleaklow peat fire, the most severe, burned for a week and destroyed about 7 km$^2$ of moorland including protected wildlife areas. Its smoke also led to the closure of major roads and disrupted air traffic at the airport of Manchester, and there were economic costs associated with restoration and suppression. The large number of spring fires was associated with a lack of winter snow cover followed by a period of cold air and strong sunshine, which prolonged the period during which grounds were frozen, causing significant damage to plants (Davies, 2008).

The greatest summer burned areas occurred in July 2006 and July 2018 (Fig.1). Both months were warmer, sunnier and drier than the average and saw several major wildfire events. Notably in 2018, the Saddleworth Moor and Winter Hill fires started in late June and burned for over three weeks, threatening properties and vital infrastructure, and damaged about 18 km$^2$ of moorland (Sibley, 2019). Other fires also occurred on defence ranges, e.g. on Salisbury Plain. The summer of 2018 was hot and dry and was preceded by cold periods in March and early April 2018, thought to have delayed the start of the growing season.

The MODIS burned areas, as annual totals, are shown in Fig.3 along with the EFFIS burned areas (available from 2008 onwards). The greatest burned areas are confirmed for 2019, 2003, 2007, 2018 and 2011. There is a good correlation between the two sources (Pearson's r = 0.82), but the EFFIS burned-area values tend to be higher than MODIS values. They have been produced using different processing methods that may allow more moderately sized wildfires to be included. Overall, no statistically significant trend was found in the MODIS data from 2003 (Mann-Kendall nonparametric trend test). A significant positive trend was identified in the EFFIS data, but little reliance can be placed on this result due to the short duration of the series.

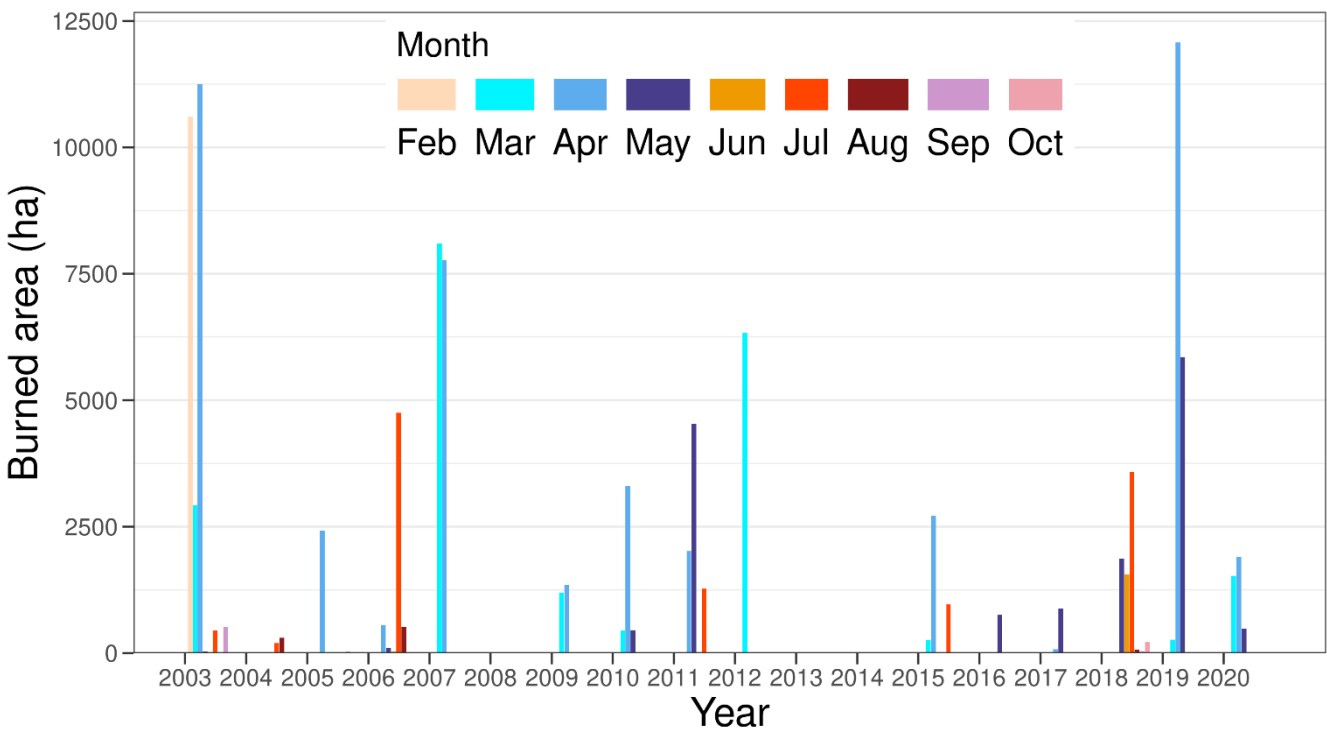

**Figure 1: Monthly burned area (km$^2$) for the UK from 2003 to 2020 from MODIS MCD64A1 data.**

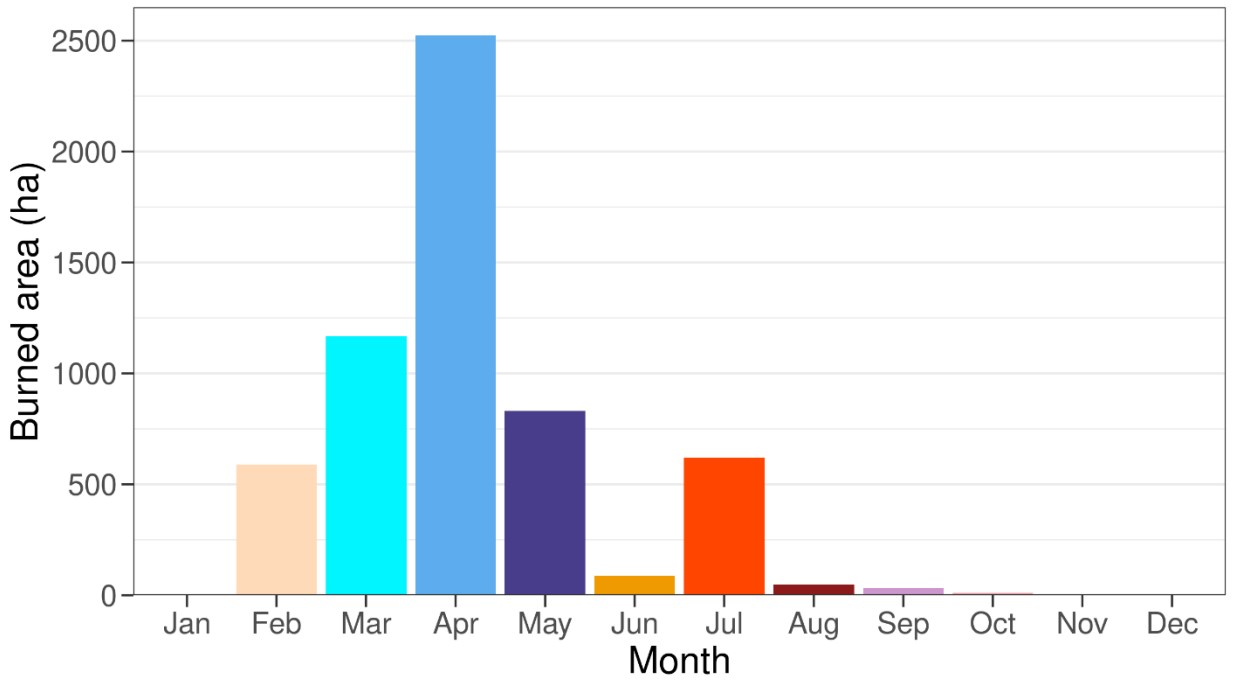

**Figure 2: Monthly mean burned area (km$^2$) for the UK from 2003 to 2020 (from MODIS MCD64A1 data product).**

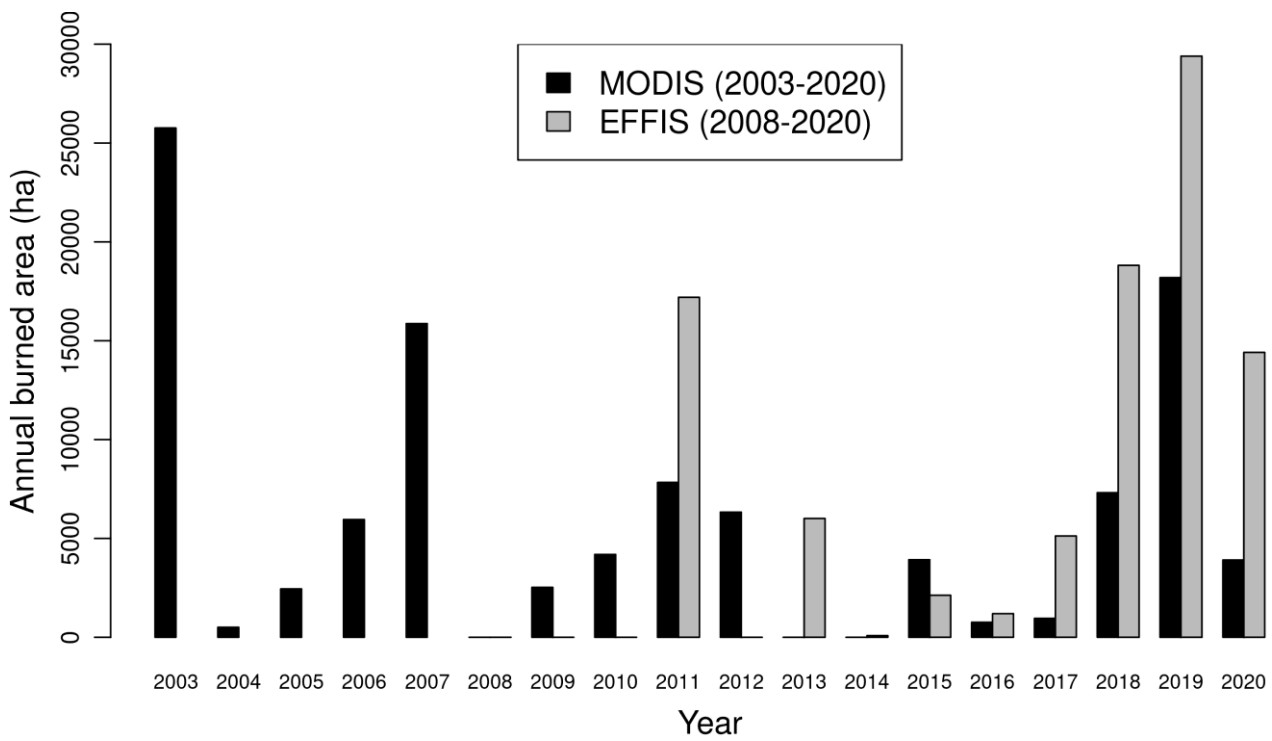

**Figure 3: Annual burned area (km$^2$) for the UK, estimated from MODIS MCD64A1 (2003–2020) and EFFIS (2008–2020).**

## 3.2 Fire weather

Although they do not represent actual wildfire events, the fire-danger indices calculated from the ERA5 reanalysis constitute a long record (1979 to date) over which to assess trend and variability of the potential for severe wildfire seasons. Spring and summer wildfire seasons are analysed separately using different fire-danger indices, as they are clearly distinct in behaviour and origins. Spring conditions are investigated using the ISI, summer conditions using the FWI (see Sect. 2.2 for further detail).

For each season, the percentage of days with dangerous fire-weather conditions was found for each grid box and these percentages were then averaged across the UK. In summer, this means any days in June, July or August with 'very high' FWI fire danger (FWI > 17.35); in spring, this means any days in March, April and May with 'high' ISI fire danger (ISI > 5). Fig.4 shows the time series from 1979 to 2020 for spring and Fig.5 the time series for summer. In spring, the most frequent occurrence of high fire danger was clearly in the most recent year of 2020, with 2011 having the next highest frequency. These were the two warmest springs in over 100 years for the UK, and were also much drier than average (Met Office, 2021). These years had a large number of wildfire events during spring, although the burned area was not as great as for 2019, which was a more average year in terms of the overall frequency of high fire danger conditions.

A non-parametric Mann-Kendall test indicated that there was no statistically significant trend, pointing to weather-related

fire conditions in spring having not changed significantly over the past 40 years despite the most recent year having the highest frequency of high fire danger. The more episodic nature of wildfires in summer is clearly visible when comparing Fig.5 to Fig. 4. The severe seasons of 1990, 1995 and 2018 (and 2006) stand out, but most years are characterised by a much smaller occurrence of very high fire danger. A Mann-Kendall test again found no evidence of a trend.

Months with 'very high' fire danger in summer were found to be closely related to the MODIS burned area (Pearson

correlation coefficient of 0.7 for June and July). For spring, there is a moderate positive relationship between burned area and fire-weather indices, with monthly correlation coefficients ranging from 0.3 to 0.6. While severe wildfire seasons generally occur when the fire danger is higher than normal, 'high' fire danger does not automatically imply the occurrence of wildfires because fire-danger indices only assess how difficult a wildfire would be to control, should one be ignited. In other words, if the fire danger is severe, fires, if they start, are likely to have a much bigger impact.

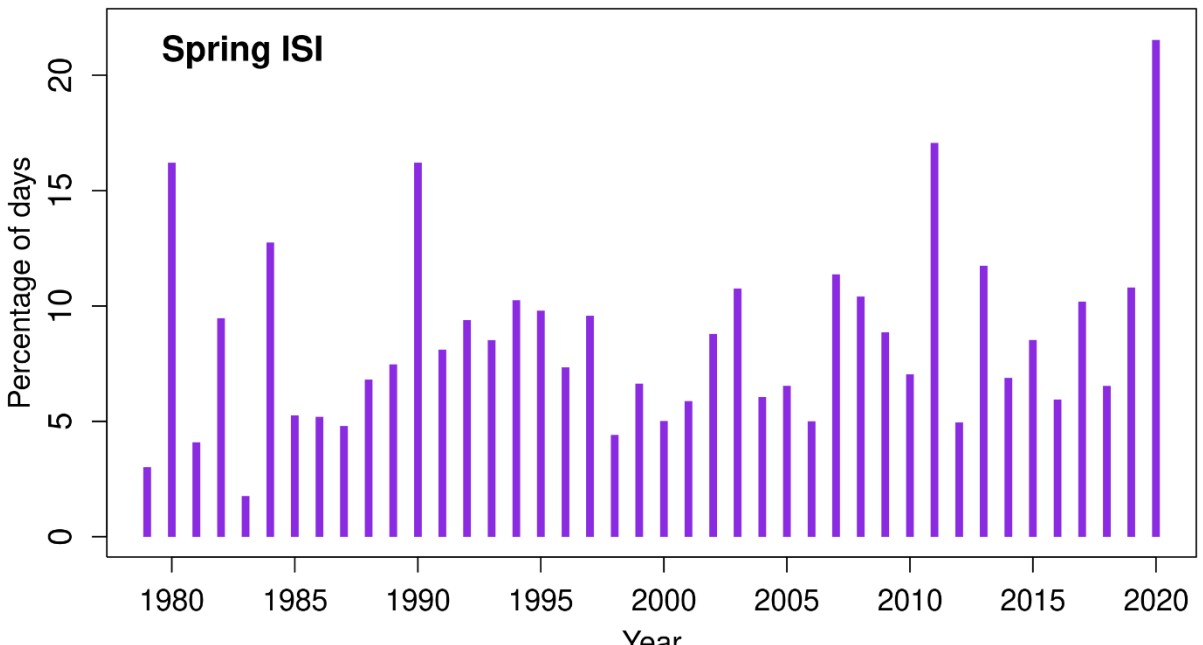

**Figure 4: Annual percentage of spring (March-April-May) days, 1979–2020, averaged over the UK, with 'high' fire danger (ISI > 5), based on ERA5 reanalysis data.**

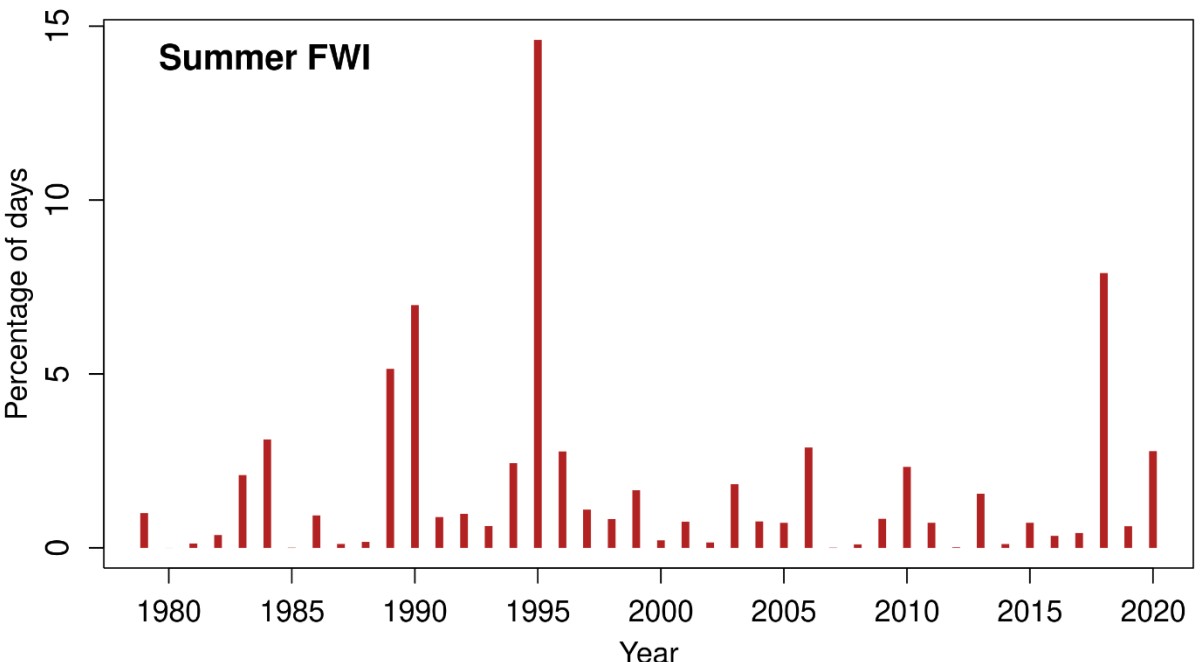

**Figure 5: Annual percentage of summer (June-July-August) days, 1979–2020, averaged over the UK, with 'very high' fire danger (FWI > 17.35), based on ERA5 reanalysis data.**

## 4 Future projections of fire weather

### 4.1 Fire danger levels

Figures 6 and 7 summarise the modelled levels of fire danger in spring and summer for England, Northern Ireland, Scotland
and Wales, during the historical period and in a 2°C and 4°C world. The plots are based on the daily 95[th] percentile index values, meaning that 5% of the area of the country is affected by higher danger levels. As before, the ISI is used to represent fire-weather conditions in spring and the FWI in summer, using the danger-class thresholds shown in Table 1. However, the 'exceptional' FWI class (FWI > 52.36) has been encompassed into a broader 'extreme' class start at FWI > 38, as the 'exceptional' danger level is – by design – very rarely reached in the data.

In spring, a slight increase in moderate and high levels of modelled fire danger is projected for each of the UK countries (Fig.6). There is a corresponding slight projected reduction in the frequency of low fire danger. The change is most marked in England, where the frequency of 'high' fire danger covering at least 5% of the country is projected to increase from 6% of days for the historical period of 1981–2010 to 9% at the 2°C global warming level and 12% at the 4°C level.

In summer, modelled fire danger increases markedly between the reference period and the 2°C global warming level, with
320 large increases projected for the 4°C level (Fig.7). In England and Wales, the percentage of days with at least 5% of the country at the 'very high' or 'extreme' danger levels is projected to double between the reference period and the 2°C global

warming level, and to increase by five times the reference frequency at the 4°C level, when the frequency reaches 46% of summer days for England. In Scotland and Northern Ireland, there is a large projected increase in the frequency of 'high' fire danger. 'Very high' danger levels remain rare but are projected to start to occur more frequently over at least 5% of the country by the 4°C level. The increase in the percentage frequency of the 'high' or 'very high' levels is less than doubling for the 2°C level and approximately four-fold for the 4°C level.

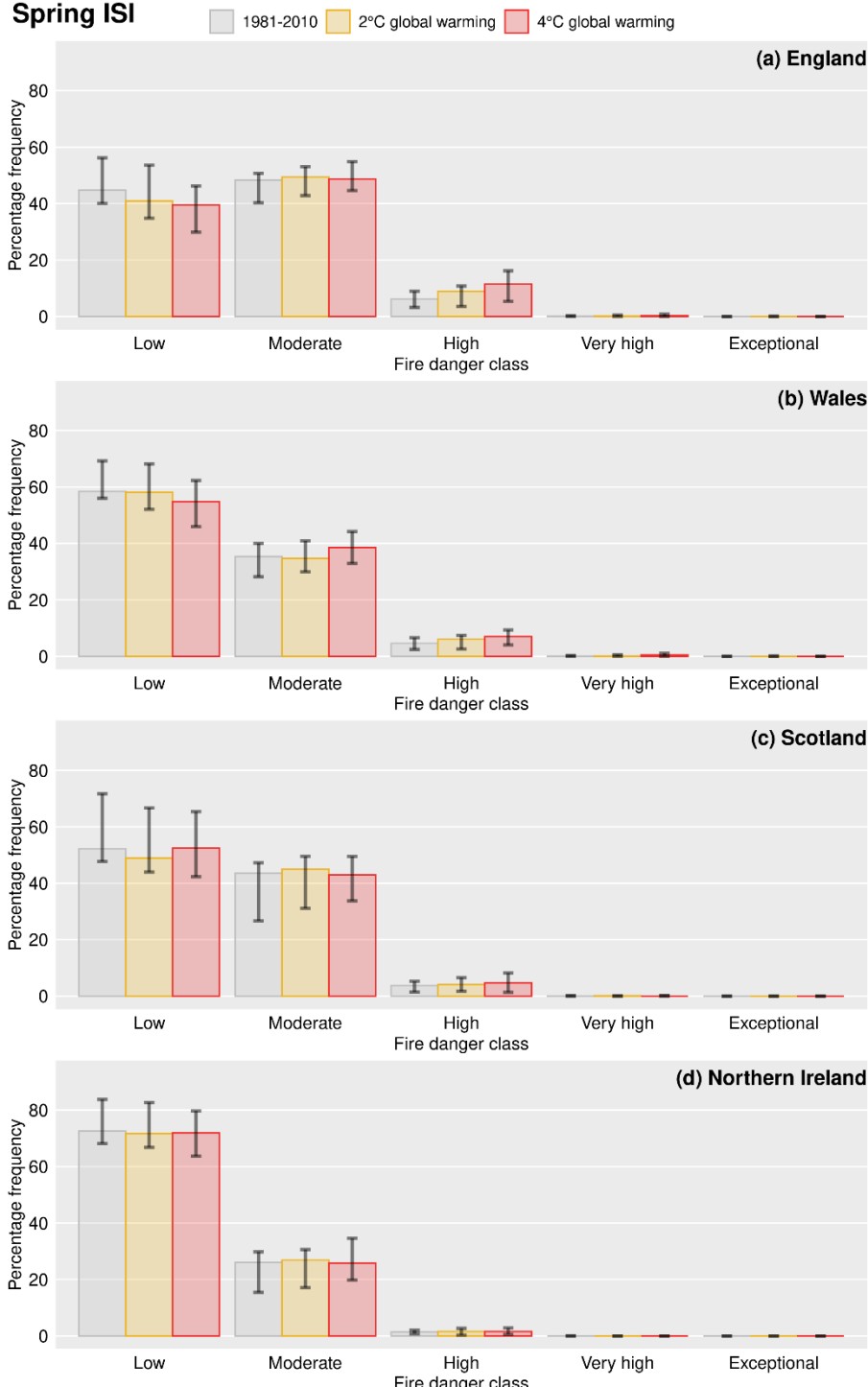

**Figure 6: Modelled percentage frequency of spring fire-danger classes based on the daily 95th percentile ISI across each country – a) England, b) Wales, c) Scotland, d) Northern Ireland. Historical period (1981–2010), 2°C and 4°C global warming levels. Coloured bars show the median of the 12 ensemble members, and the error bars show the 10th–90th percentile range.**

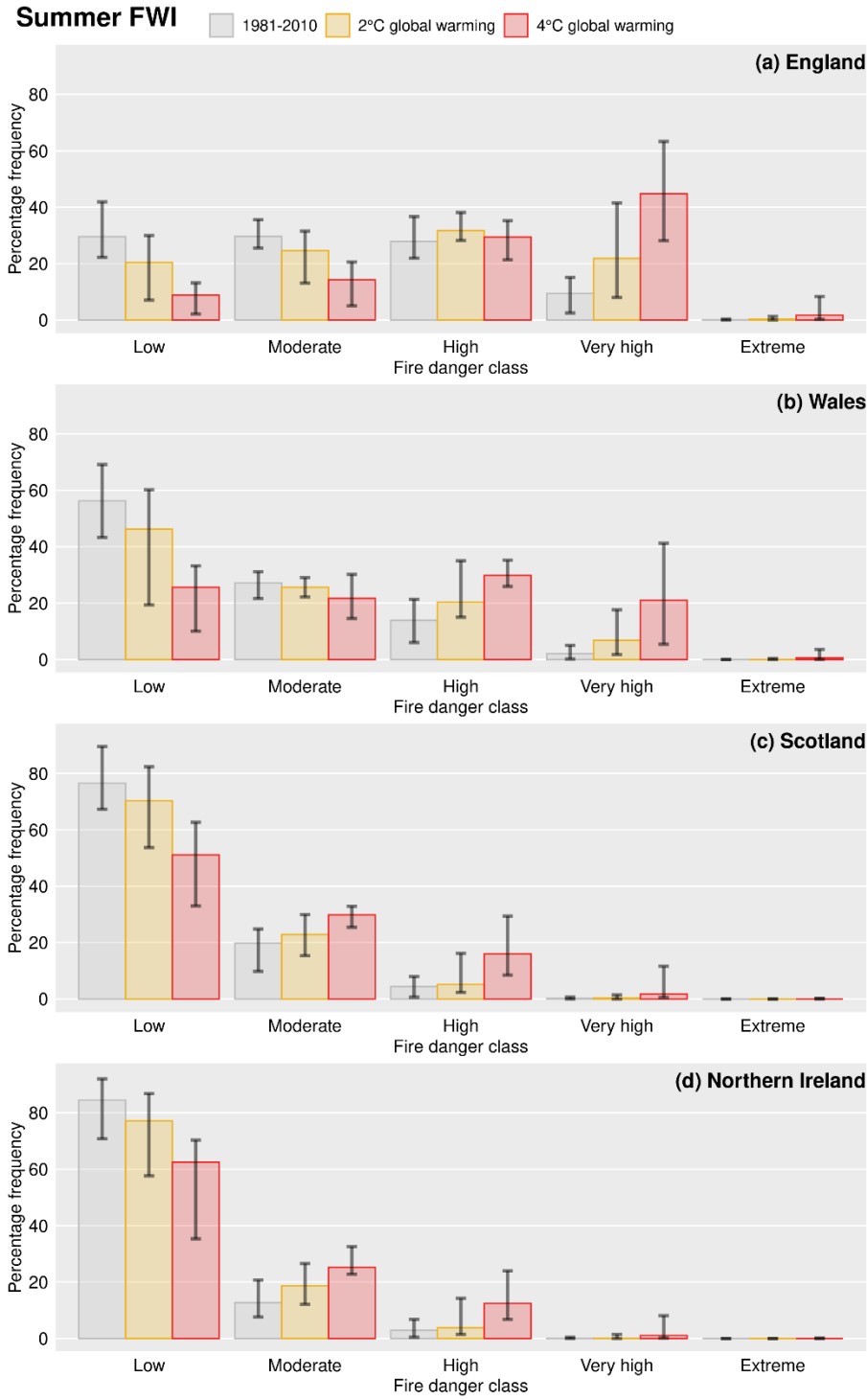

**Figure 7:** As Fig. 6, but for summer fire-danger classes based on the daily 95[th] percentile FWI. The 'exceptional' danger level has been encompassed into a broader 'extreme' danger level with FWI > 38.

## 4.2 Monthly changes

Table 2 shows the results broken down by month from February to October, extending the previous results for spring and summer to include the winter month of February and the autumn months of September and October. The modelled percentage of days with 'very high' FWI across at least 5% of the UK is projected to at least double for the 2°C global warming level for all months from June to October, and to be multiplied by at least five times by the 4°C global warming level for these months. Particularly large increases are projected for September, which could indicate an extension of the

wildfire season into the autumn, a season when previously few wildfires have occurred in the UK, although this is dependent on fuel availability. The ISI is a more useful index for the spring and late winter, and Table 2 shows no increase in the frequency of 'high' fire danger in February. Slight increases are shown for March and April, with a larger increase for May. These results highlight the importance of meeting the Paris Agreement targets, as there is a large difference between the fire risk in the projections for a 2°C world compared to a 4°C world. Even in a 2°C world, however, the results show a marked

increase in dangerous fire weather in some areas, highlighting the benefits of taking action which would keep global warming below 2°C.

**Table 2: Spatial 95th percentile across the UK of the modelled monthly percentage of days with dangerous fire weather ('high' ISI (> 5) from February to May, 'very high' FWI (> 17.35) from June to October). Median and (in brackets) the 10th–90th percentile uncertainty range from the 12 ensemble members. Values are quoted to 2 significant figures.**

| Month | 1981–2010 | 2°C global warming | 4°C global warming |
|---|---|---|---|
| February | 1.8 (0.2–2.9) | 1.5 (0.0–4.4) | 1.5 (0.0–3.0) |
| March | 2.8 (0.5–4.2) | 3.0 (0.9–6.5) | 3.0 (0.9–7.5) |
| April | 5.8 (3.0–8.4) | 6.4 (2.5–11) | 8.2 (1.6–13) |
| May | 8.4 (3.4–15) | 14 (4.4–20) | 18 (9.8–30) |
| June | 3.3 (0.3–6.0) | 8.0 (1.2–24) | 23 (7.0–34) |
| July | 9.2 (2.4–17) | 27 (6.3–44) | 44 (21–76) |
| August | 9.1 (1.8–19) | 25 (7.4–52) | 55 (36–86) |
| September | 3.2 (0.3–10) | 18 (5.5–42) | 44 (19–69) |
| October | 0.2 (0.0–2.0) | 1.1 (0.3–2.7) | 6.4 (1.9–18) |

## 4.3 Spatial changes

Spatially, the frequency of severe fire weather is not uniform across the UK. Figure 8 shows the percentage of days across the UK with dangerous fire weather in spring or summer, based on the historical ERA5 reanalysis data. In spring, 'high' fire-danger levels based on the ISI are most frequent over eastern, central and southern England. Elevated frequencies also found for coastal parts of northern England, Scotland and Wales due to higher wind speeds, especially for more exposed coastlines. In summer, dangerous fire weather is found most frequently for central and southern England. However, a bias is visible between the ERA5 data and the UKCP18 climate model data for the historical period shown in Figs. 8a and 9a, with fewer days of 'high' fire danger in the latter for the spring. The bias is much lower for the summer FWI, comparing Figs. 8b and 10a.

Projections for the future point towards a gradual strengthening of severe fire conditions in the same regions rather than a drastic change in geographical patterns. In summer, a stronger increase is visible in Fig. 10 with values across much of England ranging from 5 to 20% in a 2°C world and from 15 to 40% in a 4°C world. The projected changes are particularly dramatic in central and southern England and south Wales, and especially for the 4°C global warming level, again emphasizing the benefits of keeping global warming below 2°C. The highest frequencies for each time period are found in southeast England.

The number of days with severe fire danger is projected to remain relatively low in northern England and Scotland, but still with marked projected increases for the future periods. However, many of the most significant historical wildfire events have taken place in these northern regions. It is likely that these areas are more vulnerable to wildfire due to their more rural nature and increased remoteness, which leads to greater fuel availability and increases the risk of fires being able to spread before they can be suppressed. The spatial distribution of different types of vegetation fuels also affects vulnerability to levels of the different fire-weather indices (de Jong et al., 2016).

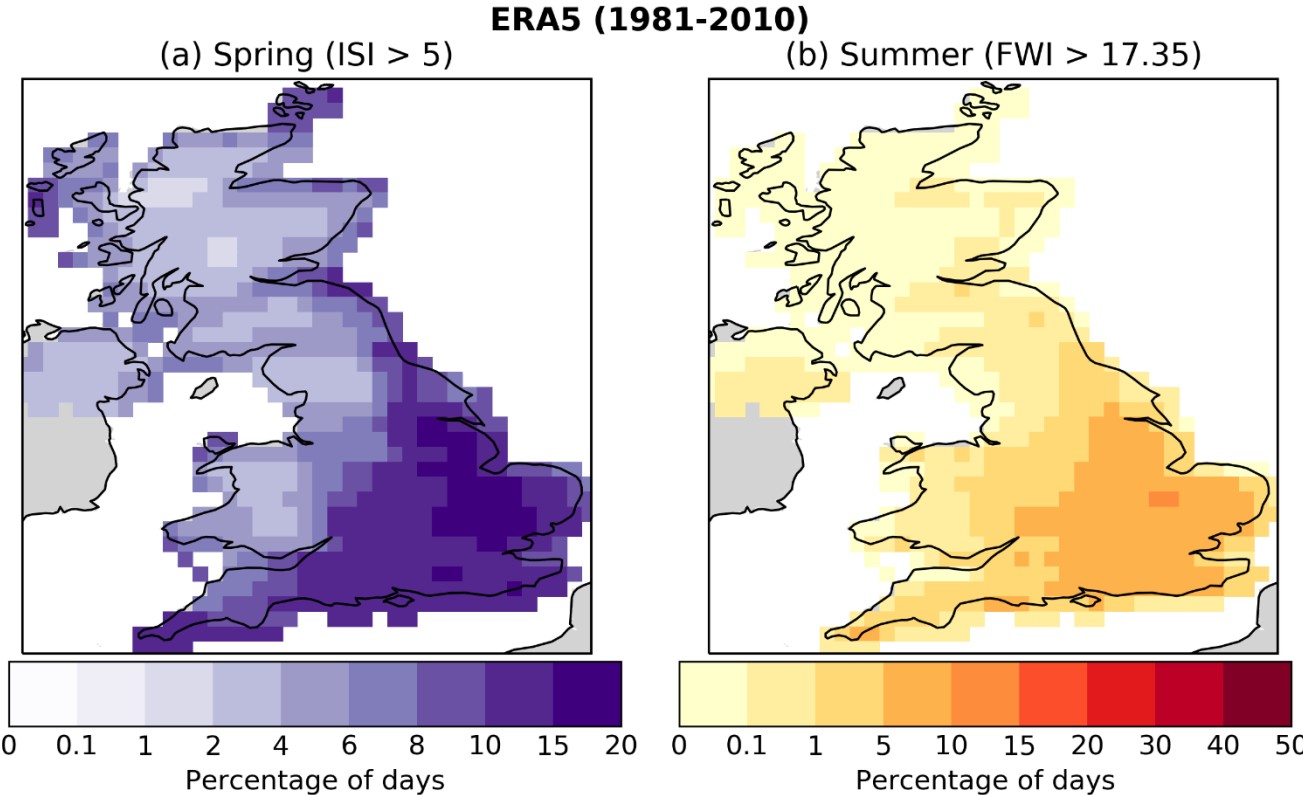

**Figure 8: Percentage of days with dangerous fire weather for a) spring (ISI > 5; at least 'high'); b) summer (FWI > 17.35; at least 'very high'). Based on the ERA5 reanalysis data for the 1981–2010 period.**

## Spring ISI (UKCP18)

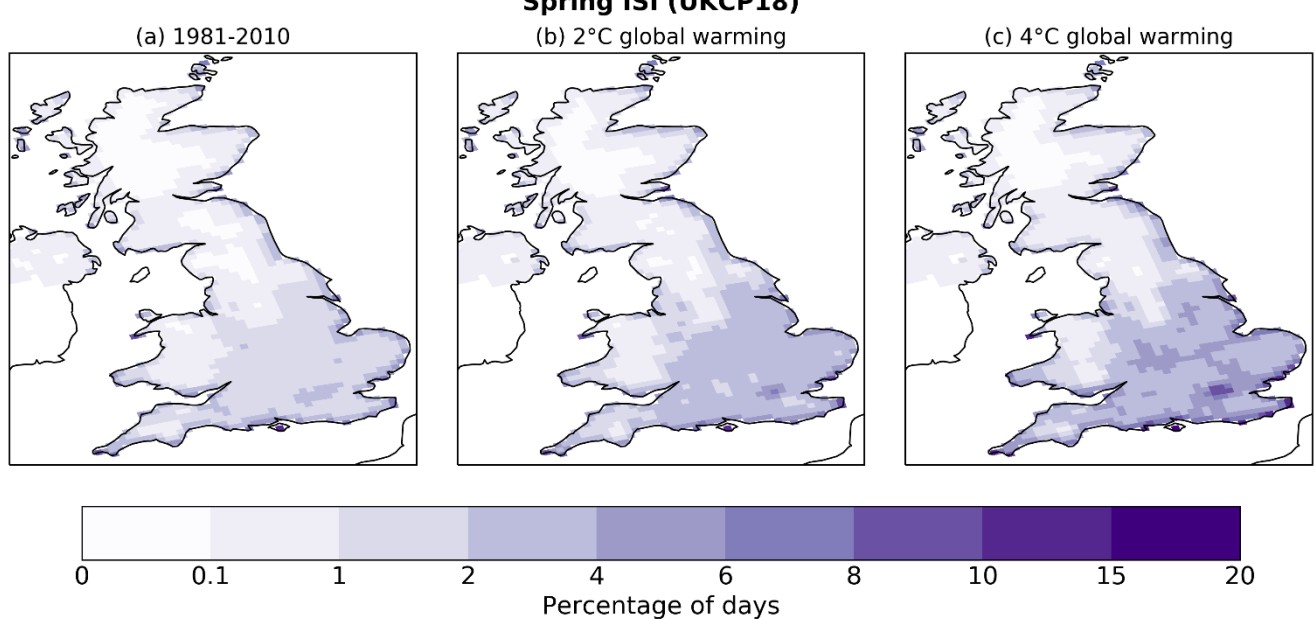

**Figure 9: Percentage of spring days with 'high' fire danger (ISI > 5) over the UK, based on the UKCP18 12 km data averaged over the 12 ensemble members. a) Historical period (1981–2010), b) 2°C global warming, c) 4°C global warming. The colour bar is the same as in Fig. 8a.**

## Summer FWI (UKCP18)

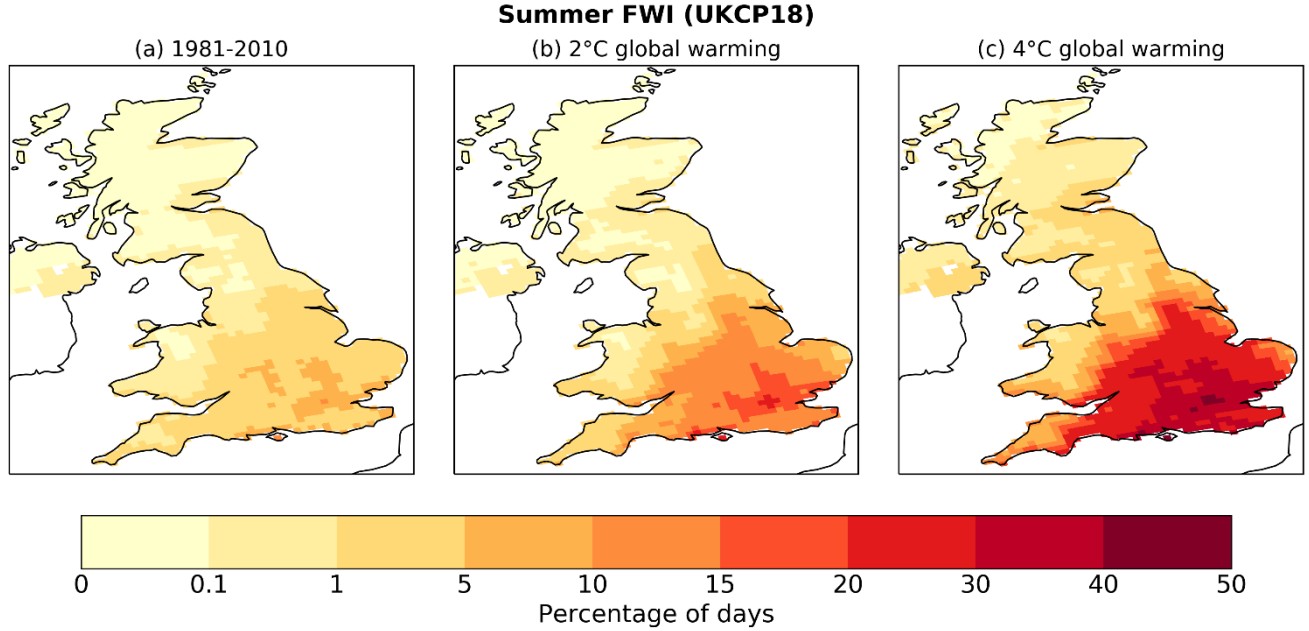

**Figure 10: Percentage of summer days with 'very high' fire danger (FWI > 17.35) over the UK, based on the UKCP18 12 km data averaged over the 12 ensemble members. a) Historical period (1981–2010), b) 2°C global warming, c) 4°C global warming. The colour bar is the same as in Fig. 8b.**

#### 4.4 Weather drivers

Table 3 shows changes in the four weather variables which contribute to the calculation of the fire weather index, using the same methods as for changes in the fire weather indices (see Section 2.4). Increases in daily maximum temperature and decreases in relative humidity are higher in summer than in spring, and at the 4°C global warming level compared to the 2°C level. For precipitation, an extra month before the season considered has been included to account for the antecedent rainfall component of the indices. Precipitation is projected to increase slightly from February to May but to decrease from May to August. Slight decreases in wind speed are projected, with again a greater change shown for summer.

Table 3: Weather variables averaged over the UK for a) spring and b) summer; historical values for the 1981-2010 period and absolute changes from 1981-2010 to the 2°C and 4°C global warming levels, showing the mean and range of 12 ensemble members from the UKCP18 12 km model

| Weather variable | 1981-2010 | 2°C GWL change | 4°C GWL change |
|---|---|---|---|
| a)  Spring | | | |
| Daily maximum temperature (°C) | 9.6 | 1.2 (0.6 to 1.7) | 2.5 (1.8 to 3.3) |
| Relative humidity (%) | 80.2 | -0.9 (-1.8 to -0.1) | -1.4 (-2.8 to -0.7) |
| Surface wind speed (m/s) | 4.6 | 0.0 (-0.2 to 0.2) | 0.0 (-0.3 to 0.1) |
| Precipitation (mm/day) | 3.5 | 0.1 (-0.2 to 0.3) | 0.2 (-0.1 to 0.6) |
| b)  Summer | | | |
| Daily maximum temperature (°C) | 17.6 | 1.9 (0.8 to 2.8) | 4.4 (3.3 to 5.5) |
| Relative humidity (%) | 78.6 | -1.8 (-3.5 to -0.8) | -4.2 (-6.6 to -2.4) |
| Surface wind speed (m/s) | 3.9 | -0.1 (-0.2 to 0.1) | -0.2 (-0.3 to -0.1) |
| Precipitation (mm/day) | 2.9 | -0.1 (-0.5 to 0.0) | -0.5 (-1.0 to -0.3) |

Changes in the weather variables were compared to the changes in the fire weather indices shown in Fig. 6 and Fig. 7, for individual ensemble members, by country and period (the 2°C and 4°C global warming levels compared to the 1981-2010 period). This revealed a strong negative relationship between change in relative humidity and change in the percentage of days with very high fire weather index in summer (see Fig. 11). The $R^2$ value from a linear model including only relative humidity is 0.76, compared with 0.84 when including all four weather variables, and no more than 0.49 when relative humidity is omitted as a coefficient (see Table A1). Summer increases in temperature and decreases in precipitation are both correlated with decreases in relative humidity, and both also have an effect on the projected increases in dangerous fire weather, but decreasing relative humidity appears to have the biggest influence. The factors influencing the smaller changes in fire weather projected for spring are less clear, with an $R^2$ of 0.53, with relative humidity again the most significant coefficient, followed by temperature (see Table A1). The slight projected increase in precipitation and the lower changes in

temperature and relative humidity compared to summer are likely to explain the lower projected increases in weather-related
fire danger in spring.

Willett et al. (2020) showed that relative humidity has decreased over recent decades for mid-latitude oceanic regions including around the UK. Warmer temperatures and lower relative humidity increase evaporative demand in the atmosphere, which increases evapotranspiration, depletes soil moisture and increases plant water stress, and this can lead to agricultural and ecological drought (Douville et al., 2021). Agricultural and ecological drought is associated with favourable conditions
for fire spread as represented in elevated values of the FWI, especially in summer for the UK.

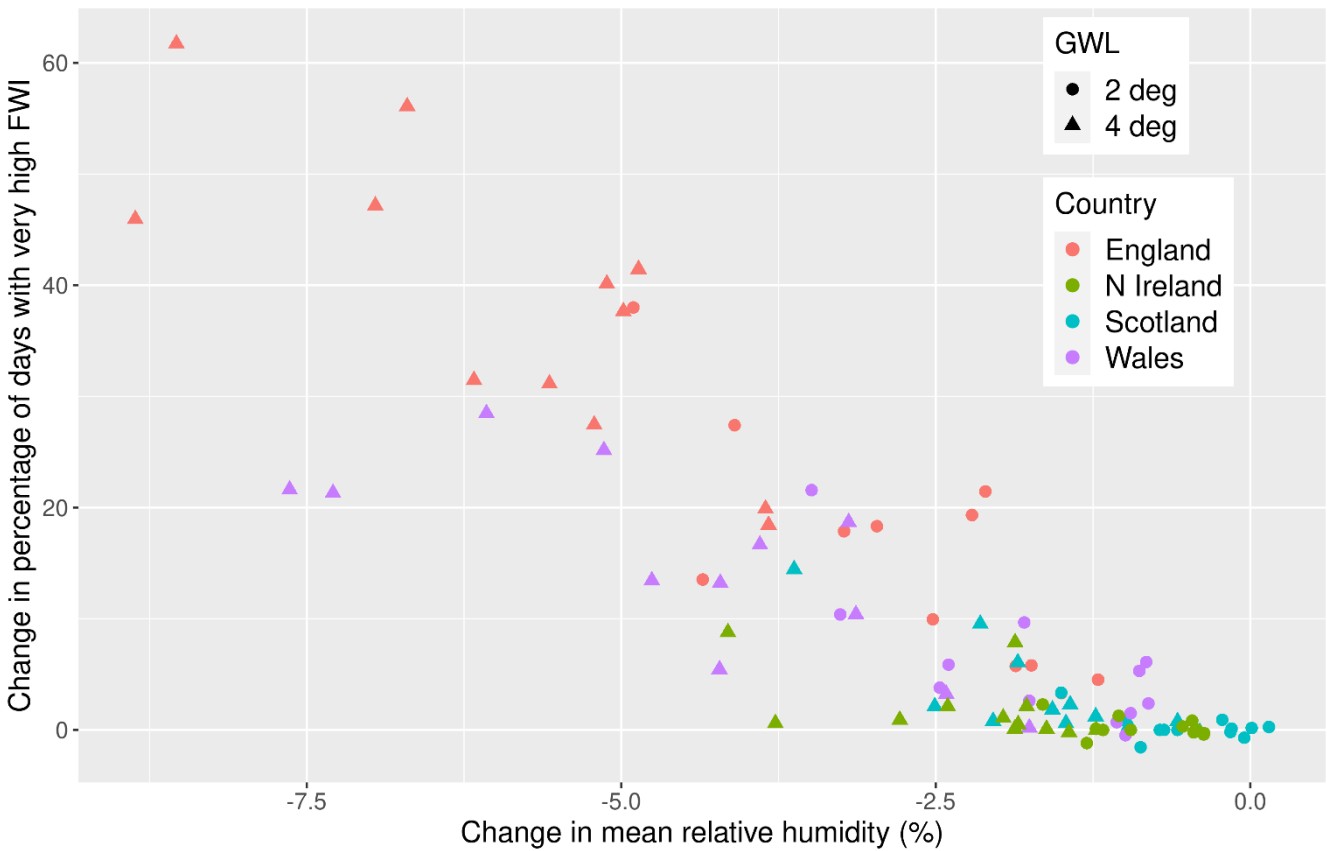

**Figure 11: Scatter plot of change in the percentage of summer days with very high FWI against change in mean relative humidity (%), by country and ensemble member. Changes are from 1981-2010 to the 2°C and 4°C global warming levels.**

**5 Discussion**

Historical variability and trend in the occurrence of wildfires in the UK have been assessed using satellite-derived burned area data over the past 17 years (2003 to 2019) and reanalysis-derived fire-danger indices over the past 40 years (1979 to 2019). In the UK, the main fire season is the spring, with the month of April having the greatest monthly total burned area across the record. This is due to a combination of a greater availability of fuel (dry and/or dead vegetation) and conducive

weather conditions (warmer, drier weather). Summer wildfires have been more episodic in nature, with most severe wildfire events being concentrated in a few hot, dry summers. No trend in historical burned area could be established, even though the past three years saw major wildfire incidents, and looking at historical wildfire-prone weather conditions, no significant trend could be identified either. The lack of a clear trend in summer may be due to the influence of rainfall (one of the weather variables used in the calculation of the fire-danger indices), which has high interannual variability and has seen a slightly increasing trend over the UK in recent decades (Met Office, 2021). The increasing trend in temperature may have been balanced by this change in rainfall. Regions most affected by 'high' fire danger in spring are most of central and southern England, coastal parts of northern England, Scotland, and Wales; in summer, hazardous fire weather is mainly concentrated in most of central and southern England.

The future projections presented here only provide a partial, weather-based view of the current and future risk of severe fire danger in the UK. Indeed, the fire-danger indices used are calculated purely from weather variables (temperature, relative humidity, wind and precipitation) and do not account for other factors such as ignition, vegetation abundance, vegetation type, land management or fire suppression. The UK population continues to grow gradually, and this could lead to an increase in ignitions, especially as climate change may lead to conditions which are more frequently suitable for outdoor activities (Belcher et al., 2021). Large climate-driven changes in land use are not expected in the UK, as land cover is dominated by direct human influence. A few studies have attempted to model future changes in wildfire due to socioeconomic and vegetation effects as well as climate at a global or continental scale, but variations in their results suggest that future changes in socioeconomic and vegetation effects on wildfire are very uncertain. However, the models agree that climate is expected to be the main driver of changing risk for northern Europe (Wu et al., 2015; Wu et al., 2021). One model also showed a significant effect of increased atmospheric $CO_2$ concentrations on increased fuel load and thus burned area (Wu et al., 2015). Indeed, important feedbacks link a changing climate, fuels and fire regimes (Davies et al., 2008). The warmer, wetter winters projected for the UK may lead to increased production of vegetation, and therefore increased fuel availability when allowed to dry during periods of elevated fire danger in spring. In summer, increasing periods of hot, dry weather may limit vegetation growth. In terms of spatial patterns, further research is needed on variations in socioeconomic and vegetation vulnerability to move from hazard mapping to risk mapping.

Our results show a projected doubling of the frequency of 'very high' fire danger levels in summer for England and Wales at the 2°C warming level and a five-fold increase at 4°C of global warming. Smaller increases are projected for spring, with a 150% increase in 'high' fire danger for England at 2°C of global warming and a doubling at 4°C. These results have important policy implications both for mitigation and adaptation. Belcher et al. (2021) provide a detailed assessment of adaptation responses. Projected changes to wildfire risk will need to be recognised and incorporated into land management and design plans for a range of land uses, including new developments at the rural-urban interface and potential re-wilding, afforestation or peatland restoration schemes aimed at helping to achieve net-zero carbon emissions. In fire-prone environments, an increased capacity to manage fuel loads both through prescribed burning and mechanical fuel removal could help to reduce the danger posed by increasingly extreme fire weather conditions. There is also a need for Fire and

Rescue Services to plan for long-term increases in the training and resources required to suppress wildfires. In order to reduce ignitions, increased social understanding of wildfire is required, and public communication of periods of high fire risk is recommended through an appropriate fire danger rating system. Lessons can be learned from the example of countries like Australia, where severe events have led to the loss of life and property, leading to the development of a new fire danger rating system to support better decision-making from emergency responders and public communication of risk and actions required (Matthews et al., 2019).

The particular set of 12 km regional projections analysed does not cover the full range of uncertainty and samples the warmer end of the climate response to greenhouse gases. We examined projected changes in fire weather at the global warming levels of 2°C and 4°C, driven by the very high emissions scenario RCP8.5. Although projections with this scenario reach GWLs of 2°C and 4°C earlier than would be reached with emissions consistent with current policies, the use of GWLs allows such changes to be considered representative of the regional climate state reached at the same level of warming but translated to a later time-period.

The UKCP18 model data have not been recalibrated (e.g. using bias-adjustment techniques) in this study. Comparison of the historical occurrence of severe fire danger from the ERA5 reanalysis (a good proxy for observations) with that from the climate model ensemble shows that biases are present, especially for spring. The large biases seen in spring mean that results for this season in particular should be interpreted with caution. However, future trends such as an increasing occurrence of severe fire danger are likely to be robust.

**6 Conclusions**

Our results suggest that wildfire can be considered an 'emergent risk' for the UK, as past events have not had widespread major impacts, but this could change in the future.

The distribution of fire-danger levels in summer revealed an increase in the frequency of 'very high' fire-danger levels being exceeded for England and Wales in a 2°C world, but a particularly dramatic increase in a 4°C world. For Scotland and Northern Ireland, large projected increases in 'moderate' to 'high' fire danger were shown. A corresponding decrease in occurrence of days with 'low' fire danger across the UK is projected. Spatially, the increase in the occurrence of severe fire weather is more pronounced in central and southern England, and in Wales.

Overall, while the occurrence of 'high' fire danger levels in spring is not projected to change as much throughout the 21st century, the spring wildfire season is already the most frequently severe season of the historical period. Combined with a projected increased fire danger in summer, and possibly the autumn, the UK could see overall more sustained severe fire danger throughout the year. This is likely to require considerable adaptation planning and policy measures, such as increased resources for fire and rescue services, and management of land to reduce the risk of ignitions and to reduce fuel availability in high-risk areas.

The large increase in wildfire hazard between the 2°C and 4°C global warming levels emphasises the importance of meeting
the targets set by the Paris Agreement in order to keep global warming levels below 2°C and so to avoid the worst impacts of
the increased risk of more frequent and intense wildfires in the UK and an extended season of hazardous fire-weather
conditions.

## Appendix A

**Table A1: Multiple $R^2$ values for linear models explaining changes in the percentage of days with high fire danger by ensemble member, country and future period with changes in different combinations of the weather variables mean relative humidity (RH), precipitation amount (pr), daily maximum temperature (tx) and surface wind speed (wind) in spring and summer.**

| Weather variables | $R^2$ (spring) | $R^2$ (summer) |
|---|---|---|
| All | 0.53 | 0.84 |
| RH + pr + tx | 0.52 | 0.83 |
| RH + pr + wind | 0.49 | 0.84 |
| RH + tx + wind | 0.53 | 0.80 |
| RH + pr | 0.49 | 0.83 |
| RH + tx | 0.52 | 0.78 |
| RH + wind | 0.46 | 0.80 |
| RH | 0.46 | 0.76 |
| tx + pr + wind | 0.32 | 0.49 |
| tx + pr | 0.22 | 0.40 |
| tx + wind | 0.23 | 0.49 |
| pr + wind | 0.02 | 0.24 |
| pr | 0.00 | 0.21 |
| tx | 0.19 | 0.39 |
| wind | 0.01 | 0.02 |

## Code and data availability

The results have been processed from three main publicly available data sources: MODIS MCD64A1 satellite-derived
burned area (Giglio et al., 2018) accessed from AppEEARS https://lpdaacsvc.cr.usgs.gov/appeears/, historical data of fire
danger indices from the Copernicus Emergency Management Service (Copernicus, 2019) accessed from the Climate Data

Store https://cds.climate.copernicus.eu/ and UKCP18 regional climate model projections (Met Office Hadley Centre, 2018). The code used to analyse these datasets to produce the figures and results presented in this article and the processed data required to reproduce the plots is available on request.

**Author contribution**

MP and EV developed the methods and code to analyse the historical data and model projections, produced the results and plots, and wrote a first draft of the manuscript. RB conceptualised the research and contributed to the writing. EP reviewed the manuscript and contributed to the writing.

**Competing interests**

The authors declare that they have no conflict of interest.

**Acknowledgements**

This work was supported by the Met Office Hadley Centre Climate Programme funded by BEIS and Defra. The authors wish to acknowledge the work of Ben Snowball in developing code to derive fire weather indices from UKCP18 model outputs.

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
