# Peer review of "Past and future trends in fire weather for the UK"

_Natural Hazards and Earth System Sciences, 2021_

## Author Response (AR1)

**Referee 1**

**Major comments**

In my view, there is currently one major flaw in the authors' work. This is relevant to the absence of any attempt to relate changes seen/projected in fire weather to changes in weather. For instance, what is driving the deterioration of fire weather conditions in the UK? Is it the wind? Is it temperature? I strongly advise the authors to try to identify the drivers of the increases seen in the indices. This would improve significantly the added-value of their work. For instance, they could use statistical methods, such as GAMs or neural networks, to attribute changes seen in the indices to changes in the basic meteorological variables.

We would like to thank the reviewer for their helpful comments. We agree that adding an assessment of the weather drivers influencing the projected changes in fire weather indices would enhance the paper. We have carried out further analysis to calculate changes in the contributing weather parameters between the periods used in the study and to investigate the relationship with changes in frequency of dangerous fire weather, for ensemble members and countries at the 2 and 4 degree levels. Relating these changes to changes in the fire weather indices indicates that decreased relative humidity is the primary factor influencing the large increase projected in very high FWI in summer. The main results from this analysis have been included in a new section 4.4 'Weather drivers' together with discussion and references.

**Minor comments**

Sect. 1.2: I think that the title of this section is misleading. By reading "Causes of wildfire", I expected to read more on ignition sources (e.g., arson, lightning). Instead, this section focuses on what we often call the fire environment (fuels, weather, topography) and how it influences wildfire activity. Consider revising the title (e.g., "Drivers of wildfire activity"?)

We have revised the title of this section as suggested

L65-69: This paragraph does not realy fit in this place. I would suggest adding a new section describing the concept of fire danger in more detail. I believe this is necessary, since this study focuses on fire danger.

Thanks for this suggestion, which we have implemented by moving this paragraph to form a new section 1.3 on Fire danger. We have also added further details to this section and a link to section 2.2

L155-156: A reference and some notes on how the thresholds were adapted for the UK would be highly welcome.

A reference has been added to a Met Office report (Kitchen, 2010) on the class thresholds, and additional text has been added to summarise the main methods used to derive the thresholds.

L155-166: I believe that some references would enhance the manuscript, along with maybe more details on how the Canadian FWI system was adapted to the pyric environment of the UK. Simply saying, for eaxmple, that FFMC is a good indicator for the UK in spring does not really say much to the interested reader. More information would be appreciated.

The explanation of the use of ISI for spring and FWI for summer has been expanded and clarified, adding reference to the results of studies by de Jong et al. (2016) and Davies and Legg (2016) which support this. References have also been added for the implementation of the Canadian FWI system for the UK (Met Office, 2005) and the Natural Hazards Partnership (Hemingway and Gunawan, 2018).

Sect. 2.3: It is not really clear if FWI was computed from the ERA5 data or the authors employed the readily available ERA5-based FWI dataset. If the first option was followed, why the use of the readily-available ERA5 FWI dataset was excluded?

The readily available ERA5-based FWI dataset was used. This section has been re-worded in the final manuscript to clarify this, and a reference added (Vitolo et al., 2020).

Figures 1-3: Burnt area should be reported in ha.

We have converted Figures 1-3 to units of ha.

**Technical corrections**

L132: "was calculated".

This sentence has been revised as suggested to: "small and spatially fragmented burned areas are not mapped at the 500 m scale at which the MCD64A1 product was calculated."

L268: Revise the sentence.

This sentence has been revised in the final manuscript to make it clearer: "In spring, the most frequent occurrence of high fire danger was clearly in the most recent year of 2020, with 2011 having the next highest frequency."

**Referee 2**

The authors reconstructed the historical fire weather in the UK and project the future changes at 2 °C and 4 °C global warming. Annual and seasonal variations in fire weather and spatial distribution of fire weather were analyzed. These were done by analyzing two fire weather indexes: ISI for spring fires and FWI for summer fires, which were calculated using historical and future climate data. The authors showed the historical patterns and projected that future "very high" fire weather in the UK would largely increase, especially in summer and in a larger warming scenario.

**Major comments**

In general, this paper was clearly and well written and provided a detailed analysis of the past and future trends in fire weather in the UK, which was not much focused on in the past. This is nice, of course. But as the authors claimed in the manuscript, fire weather does not represent actual fire events (i.e., differences in historical observed burned area and fire weather). If the authors would like to link more their results to policy suggestions on fires in the UK (this should be an important part for a regional study), more discussions about other drivers of "real" fire events will help, for example, how ignition patterns, fuel amount, land-use change, and fragmentations, suppression activities will change in the future and possible effects on fire events. I understand this is not your main focus, but I think including a separate paragraph in the Discussion regarding these other drivers will improve the broad accessibility of the paper.

We would like to thank the reviewer for their positive overall comments.

We agree that it would be beneficial to the readers of the paper to consider how other factors influencing wildfire risk may change in the future and to discuss further the policy implications of the results. We have expanded the Discussion (Section 5), adding a paragraph on the other socioeconomic and fuel drivers, including additional references to modelling studies that have been carried out on this, and a paragraph on policy implications of the results for adaptation and mitigation with further references.

**Minor comments**

L224: What do you mean by "annual cycle"? multiple-year-mean for every month?

Yes, the multiple-year mean of every month - this has been clarified in the text and in the caption of Figure 2.

L293-295: Replication with figure legend, delete.

This has been deleted as suggested

L364: Is this the 20-year mean value for panel (a)?

All 3 panels show the multi-year percentage of summer days with FWI > 17.35. The number of years for panel (a) is 30 (1981-2010), while for panels (b) and (c) it is 11 years, as described in section 2.4. This approach is used for all of the results, and we think that it is clearly explained in the figure caption and methods section.